# Analysis of Thyroid Function in ANCA-Associated Vasculitis Patients with Renal Injury

**DOI:** 10.3390/jpm14010099

**Published:** 2024-01-16

**Authors:** Wenhui Yu, Yuelan Wang, Liang Ma, Shenju Gou, Ping Fu

**Affiliations:** 1Department of Nephrology, Kidney Research Institute, West China Hospital of Sichuan University, Chengdu 610041, China; yuwenhui1@stu.scu.edu.cn (W.Y.); liang_m@scu.edu.cn (L.M.); fuping@scu.edu.cn (P.F.); 2Renal Division, Department of Medicine, Chengdu Second People’s Hospital, Chengdu 610041, China; yuelanwang@stu.scu.edu.cn; 3Department of Nephrology, West China Tianfu Hospital of Sichuan University, Chengdu 610200, China

**Keywords:** antineutrophil cytoplasmic antibody, vasculitis, renal injury, thyroid dysfunction

## Abstract

Background: Previous studies indicated common thyroid dysfunction in various kidney diseases. This study aimed to investigate the thyroid function in patients with antineutrophil cytoplasmic antibody-associated vasculitis (AAV) with renal injury. Methods: Briefly, 174 patients diagnosed as having AAV with renal injury and without previous thyroid disease history were included in the retrospective and prospective study. The clinical parameters were collected and compared between different groups. Results: Of the patients included, 24 exhibited normal thyroid function, while 150 had thyroid dysfunction, including 55 (36.67%) with hypothyroidism. Those AAV patients with thyroid dysfunction showed different clinical parameters from those with normal thyroid function. The patients were followed up for a median of 68.6 (64.3; 72.8) months. Those with thyroid dysfunction were more prone to progressing to dialysis dependence compared to the group with normal thyroid function. Logistic regression analysis showed advanced age and decreased albumin as independent risk factors for thyroid dysfunction in patients with AAV. Survival analysis and multivariate Cox regression analysis showed that thyroid dysfunction was a risk factor for AAV patients with renal injury to progress to the endpoint of dialysis dependence. Conclusion: Thyroid dysfunction, predominantly hypothyroidism, was commonly complicated in AAV patients with renal injury. AAV patients with thyroid dysfunction were presented with different clinical parameters and more prone to progressing to dialysis dependence compared to those with normal thyroid function.

## 1. Introduction

Anti-neutrophil cytoplasmic antibody (ANCA)-associated vasculitis (AAV) is characterized by necrotizing inflammation of small vessels, which consists of granulomatosis with polyangiitis (GPA), microscopic polyangiitis (MPA) and eosinophilic granulomatosis with polyangiitis (EGPA) [1,2,3]. AAV patients have a high mortality rate, especially since most deaths occur within one year of onset [4]. ANCAs are important serologic biomarkers of ANCA-associated vasculitis (AAV). Myeloperoxidase (MPO) and proteinase 3 (PR3) are two important antigens targeted by ANCA. With the improvement of the technique for inspecting ANCAs, AAV is becoming a common immune system disease in East Asia [5]. AAV may involve various organs throughout the body, with the kidneys and lungs being the most commonly affected. Renal involvement typically manifests as a progressive deterioration in renal function, and may even progress to require renal replacement therapy (RRT) in a short period of time. Furthermore, renal involvement is an important predictor of mortality in AAV [6].

Thyroid hormones have a profound impact on the metabolism, growth, development, and differentiation of tissues throughout the body. Moreover, thyroid hormones modulate various aspects of immune function, including immune cell differentiation, proliferation, and cytokine production. They also influence the balance between different types of immune cells, such as T helper 1 (Th1) and T helper 2 (Th2) cells, which play a crucial role in regulating immune responses [7].

In addition, the thyroid hormone also regulates neutrophil respiratory burst by regulating gp91phox, the catalytic core of NADPH oxidase, which is an important part of the pathogenesis of AAV [8,9]. Thyroid dysfunction can disrupt the delicate balance of the immune system, leading to an unbalanced inflammatory response [10]. A high prevalence of thyroid dysfunction has been reported in autoimmune diseases and kidney diseases, such as systemic lupus erythematosus and nephrotic syndrome [11,12]. Previous studies also reported an increased risk of thyroid dysfunction in AAV patients [13,14]. Lionaki et al. and Prendecki et al. reported that the prevalence of thyroid dysfunction in AAV patients was about 20%, which was significantly higher than its prevalence in the general global population (0.5–5.3%) [15]. In addition, compared to AAV patients without thyroid dysfunction, they found that those with thyroid dysfunction were more likely to be women and MPO-ANCA-positive [16,17]. However, the thyroid function status in AAV patients with renal injury remains unclear. Moreover, the clinical and prognostic relevance of thyroid dysfunction in AAV patients with renal injury is also unclear. This study aimed to investigate thyroid function in AAV patients with renal injury and to explore the clinical relevance of renal dysfunction in the patients.

## 2. Materials and Methods

### 2.1. Patients

This was a retrospective and prospective study that recruited 174 AAV patients with renal injury in total. Patients diagnosed as having AAV with renal injury according to the criteria of the Chapel Hill Consensus Conference 2012 [2] from November 2011 to December 2020 from West China Hospital of Sichuan University were included. The follow-up was conducted until March 2022. All patents involved in this research had completed an ANCA test and thyroid hormone test at least once during the hospitalization period. Patients who met the following criteria were excluded: (1) a lack of thyroid-related laboratory tests or medical record integrity; (2) a history of thyroid diseases before the diagnosis of AAV; (3) an age of <18 years; (4) the presence of concomitant diseases or autoimmune diseases other than AAV, such as systemic lupus erythematosus; and (5) previous radioactive iodine therapy.

The present study was approved by the Ethics Committee of West China Hospital of Sichuan University (Number 2021848). The informed consent for publication of their clinical data was obtained from all the patients or from their lineal relatives in the case where the patient had passed away. All the procedures followed the Declaration of Helsinki principles.

### 2.2. Classification

Thyroid dysfunction was defined as a deviation of thyroid hormone from the normal reference range including the levels of free triiodothyronine (FT3), free thyroxine (FT4), or thyroid-stimulating hormone (TSH). According to the level of thyroid hormone, relative subclasses of thyroid dysfunction were defined as follows: (1) subclinical hypothyroidism: an elevation in TSH level despite normal serum levels of FT3 and FT4; (2) subclinical hyperthyroidism: a decline in TSH level despite normal serum levels of FT3 and FT4; (3) hypothyroidism: low FT4 and FT3 with elevated TSH; (4) low triiodothyronine (T3): low serum levels of FT3 accompanied by normal FT4 and normal TSH levels; (5) low thyroxine (T4): low serum FT4 levels with normal FT3 levels and normal TSH levels; (6) low T3T4: normal TSH levels with low serum FT3 and low serum FT4 levels. The classification was based only on hormone levels and was not equivalent to clinical diagnosis.

### 2.3. Research Design

Demographic data were collected at the time of admission. Blood samples were collected from the peripheral vein in the morning during the first three days of hospitalization. A database was established to collect patients’ demographic information (name, gender, and age), medical history, Birmingham vasculitis activity score (BVAS), organ involvement, complications, and laboratory data including the following items: (a) blood routine examination: hemoglobin (Hb), leukocyte (WBC), neutrophil count (NEUT), neutrophilic granulocyte percentage (NEUT%), and cystatin-C; (b) biochemistry of blood: serum album (Alb), serum creatine (Cr), estimated glomerular filtration rate (eGFR), uric acid (UA), and blood urea nitrogen (BUN); (c) urine analysis: urine protein, urine erythrocyte, urine leukocyte, urinary protein creatinine ratio (PCR), urinary albumin creatinine ratio (ACR), and 24 h urine protein (PRO); (d) thyroid hormone: free triiodothyronine (FT3), free thyroxine (FT4), thyroid-stimulating hormone (TSH), triiodothyronine (T3), thyroxine (T4), reverse triiodothyronine(rT3), thyroid peroxidase antibodies (TPOAb), and thyroglobulin antibodies (TgAb). Clinical data were obtained from the Hospital Information System (HIS), and the eGFR was calculated by the Chronic Kidney Disease Epidemiology Collaboration (CKD-EPI) equation [18]. Patients were divided into the normal thyroid function and thyroid dysfunction group, referring to the levels of FT3, FT4 and TSH. According to ANCA-ELISA test, patients were divided into an ANCA-negative group, a MPO-ANCA-positive group, a PR3-ANCA-positive group, and a MPO-ANCA and PR3-ANCA double-positive group. Clinical parameters and laboratory tests between different groups were analyzed. 

### 2.4. Follow-Up

Patients were followed up from the index date until their date of death or March 2022, whichever came first. Information regarding disease progression to the stage of maintenance dialysis or death was collected through telephone inquiries. The end point was defined as dialysis dependence or death. 

### 2.5. Statistics

Continuous data with normal distribution were presented as means ± standard deviations (SDs). Non-normally distributed data were presented as the median and quartile. Categorical variables were expressed by percentages. A Kolmogorov–Smirnov test and P-P diagram were used for the normality test of the included data. Quantitative parameters were assessed with the *t*-test between groups for normally distributed data, and non-normally distributed data were assessed with the Mann–Whitney U test. Categorical data were compared using the chi-square test or Fisher’s exact test. Pearson’s correlation coefficient was used to describe the linear correlation between normally distributed data, and Spearman’s correlation coefficient was used to describe the linear correlation between non-normally distributed data. Kaplan–Meier analysis was used to analyze survival rates of the normal thyroid function group and thyroid dysfunction group and to create graphs of the observed survival curves, while the log-rank test was used to compare curves from different groups. Logistic regression was used to analyze the risk factors related to thyroid dysfunction. A two-tailed *p*-value of less than 0.05 was considered statistically significant. Statistical analyses were performed using SPSS Statistics for Windows Version 25.0 (SPSS Inc., Chicago, IL, USA).

## 3. Results

### 3.1. The General Information of Included AAV Patients with Renal Injury

Initially, 636 AAV patients with renal injury were screened from November 2011 to December 2020. After careful and precise consideration, we screened out and excluded two patients with primary hyperthyroidism and 460 patients with a lack of thyroid-related laboratory tests or medical record integrity, and 174 AAV patients were finally included in the study. In this cohort of 174 AAV patients, the mean age was 56.9 ± 16.5 years, 102 patients were females (58.6%), and the mean BVAS score was 16.97 ± 6.22. Table 1 summarized the baseline demographic and clinical characteristics of enrolled patients.

There were twenty-four patients (13.79%) in the group of AAV patients with normal thyroid function. The thyroid dysfunction group consisted of one hundred and fifty (86.21%) individuals. Among the AAV patients with thyroid dysfunction, fifty-eight cases (38.7%) were presented with low T3, fifty-five cases (36.7%) with hypothyroidism, twenty-eight cases (18.7%) with low T3 and T4, four cases with subclinical hypothyroidism, two cases with normal FT4, decreased FT3 and decreased TSH levels, one case with low T4, one case with subclinical hyperthyroidism, and one case with decreased FT3, FT4 and TSH levels. 

### 3.2. Comparison of Clinical Parameters between Normal Thyroid Function Group and Thyroid Dysfunction Group in AAV Patients with Renal Injury

The male-to-female ratio in the thyroid dysfunction group was 3:2. The AAV patients with thyroid dysfunction had a higher mean age and BVAS score compared to those with normal thyroid function (58.8 ± 18.8 years vs. 45.1 ± 16.2 years, *p* = 0.002; 17.49 ± 5.98 vs. 13.67 ± 6.77, *p* = 0.005; respectively). The levels of FT3, FT4, T3, and T4 in AAV patients with thyroid dysfunction were lower than those of patients with normal thyroid function and the level of TSH was higher than those with normal thyroid function (2.45 ± 0.73 pmol/L vs. 4.13 ± 0.40 pmol/L, *p* < 0.001; 13.58 ± 3.78 pmol/L vs. 16.86 ± 2.25 pmol/L, *p* < 0.001; 0.88 ± 0.32 nmol/L vs. 1.55 ± 0.32 nmol/L, *p* < 0.001; 69.79 ± 23.08 nmol/L vs. 101.93 ± 11.48 nmol/L, *p* = 0.003; 2.80 (1.40, 5.45) mU/L vs. 2.07 ± 1.27 mU/L, *p* = 0.008; respectively). 

The levels of hemoglobin, ALB, and eGFR in AAV patients with thyroid dysfunction were lower than those of patients with normal thyroid function (85.92 ± 20.8 g/L vs. 104.00 ± 24.93 g/L, *p* < 0.001; 33.05 ± 6.28 g/L vs. 38.86 ± 4.68 g/L, *p* = 0.003; 11.00 (6.62, 23.44) mL/min/1.73 m^2^ vs. 44.32 (13.89, 68.75) mL/min/1.73 m^2^, *p* < 0.001; respectively). On the other hand, the level of serum creatinine (CR) and the level of urea nitrogen (BUN) in AAV patients with thyroid dysfunction were higher than those of patients with normal thyroid function (457.14 ± 306.71 umol/L vs. 260.58 ± 215.55 umol/L, *p* = 0.09; 22.07 ± 13.15 mmol/L vs. 12.50 ± 6.71 mmol/L, *p* < 0.001; respectively) (Table 1). There were no significant differences in gender ratio, levels of thrombocytes, leukocyte, cystatin C, uric acid, triglyceride, and cholesterol, the urine protein creatinine ratio (PCR), TgAb, and TPOAb between AAV patients with normal thyroid function and those with thyroid dysfunction.

According to the ANCA test results, forty-five cases (25.86%) of the 174 AAV patients were ANCA-negative, one hundred and nineteen cases (68.39%) were MPO-ANCA-positive, seven cases (4.02%) were PR3-ANCA-positive, and three cases (1.72%) were MPO-ANCA and PR3-ANCA double-positive. There were no significant differences in thyroid hormone levels among groups of patients with different ANCA positivity status.

### 3.3. Relationship between Thyroid Hormone and Clinical Parameters in AAV Patients with Renal Injury

The correlation analysis between thyroid hormones and clinical parameters in AAV patients with renal injury showed that the level of FT3 was positively correlated with the levels of hemoglobin, albumin and eGFR(R = 0.42, *p* < 0.001; R = 0.51, *p* < 0.001; and R = 0.33, *p* < 0.001; respectively), while the level of FT3 was negatively correlated with age, the levels of serum creatinine, urea nitrogen, and cystatin C, the urine protein creatine ratio (PCR), and BVAS scores(R = −0.23, *p* = 0.003; R = −0.30, *p* < 0.001; R = −0.32, *p* < 0.001; R = −0.39, *p* < 0.001; R = −0.25, *p* < 0.05; R = −0.47, *p* < 0.001; respectively). The level of FT4 was positively correlated with the levels of hemoglobin, albumin and eGFR while it was negatively correlated with the levels of serum creatinine, urea nitrogen, cystatin C, uric acid, triglyceride and PCR (R = 0.22, *p* = 0.004; R = 0.21, *p* = 0.007; R = 0.33, *p* < 0.001; R = −0.29, *p* < 0.001; R = −0.32, *p* < 0.001; R = −0.34, *p* < 0.001; R = −0.17, *p* < 0.05; R = −0.23, *p* = 0.032; R = −0.27, *p* = 0.006; respectively). T3 was positively correlated with hemoglobin, albumin, and the urine album creatine ratio (ACR), while it was negatively correlated with age and BVAS scores (R = 0.42, *p* = 0.002; R = 0.44, *p* = 0.001; R = 0.90, *p* = 0.015; R = −0.096, *p* < 0.05; R = −0.34, *p* < 0.05; respectively). T4 was positively correlated with hemoglobin and albumin, while it negatively correlated with the neutrophil ratio (R = 0.30, *p* < 0.05; R = 0.35; *p* = 0.016; R = −0.42, *p* = 0.002; respectively). TSH was positively correlated with age, while it negatively correlated with the counts of leucocytes and neutrophils (R = 0.24, *p* = 0.002; R = −0.18, *p* < 0.05; R = −0.18, *p* < 0.05; respectively) (Figure 1). Logistic regression analysis showed that advanced age (OR: 1.073, 95%CI: 1.1024–1.125; *p* = 0.003) and decreased levels of albumin (OR: 0.855, 95%CI: 0.733–0.998; *p* = 0.047) were independent risk factors for thyroid dysfunction in AAV patients with renal injury.

### 3.4. Survival Analysis

In this cohort of 174 individuals, 151 patients (86.78%) were fully followed up. Briefly, 23 patients (13.21%) of the study were lost to the follow-up. Among the 151 patients included in the survival analysis, 23 patients had normal thyroid function and 128 patients had thyroid dysfunction. The median follow-up time was 68.6(64.3, 72.8) months. Kaplan–Meier survival analysis showed that the thyroid dysfunction group and normal thyroid function group in AAV patients with renal injury predicted significantly different survival curves. Furthermore, the AAV patients with thyroid dysfunction tended to have a worse kidney prognosis (*p* = 0.004, Figure 2). Among the 128 AAV patients with thyroid dysfunction, 78 patients reached the endpoint of dialysis dependence, while among the 23 AAV patients with normal thyroid function, 3 patients reached the endpoint of dialysis dependence. Univariate Cox regression analysis showed that thyroid dysfunction was a risk factor for AAV patients with renal injury to progress to the endpoint of dialysis dependence (HR: 3.921, 95%CI: 1.434–10.722, *p* = 0.008). Multivariate Cox regression analysis showed that thyroid dysfunction was also significantly associated with the risk of progression to dialysis dependence (HR: 4.051, 95%CI: 1.466–11.194, *p* = 0.007).

## 4. Discussion

Previous study revealed an increased risk of thyroid dysfunction in AAV patients [13,14,15]. In clinical practice, a subset of AAV patients with renal involvement exhibiting thyroid dysfunction were observed. However, the clinical and prognostic relevance of thyroid dysfunction in AAV patients with renal injury remained unclear. The present study analyzed the thyroid function in the cohort of AAV patients with renal injury. 

Thyroid dysfunction was not rare in AAV. The data in the present study revealed that thyroid dysfunction in AAV patients mainly manifested as hypothyroidism. In the current study, the prevalence of hypothyroidism in AAV patients was 31.6%. Prior studies reported an incidence of hypothyroidism ranging from 4% to 20% in AAV patients, and it was strongly associated with positive MPO-ANCA [15,16,17,19]. In the present study, 68.39% of the included patients were MPO-ANCA-positive, and this might contribute to the higher prevalence of thyroid dysfunction in the patients of the present study than that of patients in previous reports. However, the present data revealed no significant difference in thyroid hormone levels among patients with different ANCA subtypes. 

In the current study, the level of hemoglobin was significantly lower in AAV patients with thyroid dysfunction than that of patients with normal thyroid function. Furthermore, the levels of FT3, FT4, T4 and T3 were positively correlated with the level of hemoglobin. Anemia was a common complication in patients with ANCA-associated renal vasculitis. Wang et al. reported a retrospective study including 145 AAV patients from China; 94.4% of the patients had anemia with 80.5% of these cases being moderate or severe. Furthermore, severe anemia unparalleled with renal failure and hemoptysis was a clinical feature of AAV patients [20]. Severe anemia that was not consistent with renal failure and hemoptysis indicated that the mechanism of anemia in patients with AAV was much more complex. Low levels of hemoglobin had a negative effect on short-term prognosis in AAV patients [21]. Previous studies had suggested that the pathogenesis of anemia in AAV with renal injury differed from that of other immune-related diseases, and the factors that might contribute to the development of anemia in AAV patients have not been fully clarified yet. The reported contributing factors of anemia in AAV included renal dysfunction, alveolar hemorrhage, malnutrition, the use of immunosuppressive drugs, frequent blood sampling during hospitalization, and iron deficiency [22]. Baier et al. reported that peritubular capillaritis, Bowman’s capsule rupture and renal anemia were also the causes of anemia in patients with renal injury in ANCA-associated vasculitis [23]. However, the relationship between these factors and the severity of anemia in patients with ANCA-associated renal vasculitis has not been elucidated in detail. Omar et al. reported a higher incidence of anemia associated with thyroid dysfunction, reaching 57.1% in hypothyroidism and 40.9% in hyperthyroidism [24]. The study by Dilek et al. revealed that patients with chronic hypothyroidism could suffer from erythropoietin resistance [25]. Kjaergaard AD et al. found impaired hemoglobin synthesis in patients with autoimmune thyroid disease [26]. Thyroid hormone could directly stimulate the proliferation of erythrocyte precursors, and also promote erythropoietin production by increasing the expression of the erythropoietin gene in the kidney, and then affect hemoglobin levels [27]. The close association of thyroid hormones with the level of hemoglobulin found in the present study might provide a new clue to explain the mechanism of anemia in AAV. The downregulated thyroid hormones might be one of the causes that contributed to anemia in patients with AAV. For clinical physicians, in the differential diagnosis of anemia in AAV patients, considerations should include not only the most common causes but also thyroid dysfunction.

In the present study, the level of creatinine was higher and the level of eGFR was lower in AAV patients with thyroid dysfunction than those in patients with normal thyroid function. Correlation analysis showed that FT3 and FT4 were negatively correlated with the level of serum creatinine and positively correlated with eGFR. Similarly, previous studies found that the degree of renal dysfunction was more severe in thyroid dysfunction patients with nephrotic syndrome or systemic lupus erythematosus [11,28]. Walter Reinhardt et al. also found that the eGFR was positively correlated with FT3, FT4, T3 and T4 in patients with chronic kidney disease [29]. Accumulated research showed that thyroid dysfunction was common in patients with CKD [29,30]. However, AAV patients with kidney injury was often manifested as rapidly progressive glomerulonephritis, with a rapid deterioration of kidney function in the short term. In the present study, we found a close association between kidney insufficiency and thyroid dysfunction in AAV patients, which supported a timely link between renal dysfunction and thyroid dysfunction. The similar association found in patients with AAV and those with CKD indicated that renal impairment might be the key factor that induced thyroid dysfunction in patients with kidney disease, which could also help explain the mechanism of thyroid dysfunction in patients with AAV. Thyroid function and renal function may influence each other. Hypothyroidism may lead to renal dysfunction by reducing blood flow to the kidneys, affecting the glomerular structure and glomerular filtration rate [30]. Thyroid hormones could also influence the active transport processes in renal tubules, causing a decrease in the tubular secretion of creatinine, ultimately leading to an elevation in blood creatinine levels [31,32,33]. Renal dysfunction may cause thyroid dysfunction. Previous analysis showed that the risk of hypothyroidism increased by 18% for every reduction in eGFR of 10 mL/min/1.73 m^2^ in patients with chronic kidney disease [30]. A greater loss of urinary protein in patients with renal dysfunction could lead an increased loss of thyroid-binding protein, resulting in a decrease in thyroid hormone levels [34]. Furthermore, the retention of iodine due to impaired renal excretion has been hypothesized as a potential mechanism for thyroid dysfunction [35]. In the present study, we followed up patients and found that thyroid dysfunction was a risk factor for AAV patients with renal injury to progress to the endpoint of dialysis dependence. This finding highlighted the potential clinical implications of thyroid dysfunction on renal outcomes in AAV patients. Accumulated studies have reported that thyroid dysfunction was associated with abnormal serum creatinine levels and eGFR, and when thyroid dysfunction was reversed with treatment, renal function was also improved [36,37,38,39]. Shin et al. reported two studies targeting the chronic kidney injury population, and the results indicate that thyroid hormone therapy not only preserved renal function more effectively but also was an independent predictor of kidney prognosis in chronic kidney disease patients with subclinical hypothyroidism [40,41]. Furthermore, in a randomized, double-blind, placebo-controlled trial involving 136 individuals with early-stage type 2 diabetic nephropathy and concurrent hypothyroidism, 48 weeks of levothyroxine treatment, compared to a placebo, led to a reduction in the urinary albumin excretion rate and exerted a renal protective effect on the patients [42]. Although these limited data suggest a benefit, levothyroxine bears a narrow toxic-to-therapeutic window [43]. Furthermore, so far, there has been no research on whether or not treating hypothyroidism is beneficial for patients with ANCA-associated vasculitis. Further relevant studies are needed. For clinical physicians, it is important to realize that thyroid dysfunction, especially hypothyroidism, is common and closely related to renal dysfunction in patients with AAV. Determining whether or not to treat thyroid dysfunction in patients with AAV requires appropriate collaboration between endocrinologists and nephrologists.

Several limitations in this study should be considered. Firstly, it was a single-center study, and the data were collected retrospectively from clinical databases. The screening for thyroid function was decided by physicians. There might be a possible bias that the included patients might be suspected of having thyroid dysfunction. However, we checked all the medical records and found that all patients included had no history of previous thyroid disease. There were many AAV patients who did not have thyroid function data because different physicians had different habits of screening for thyroid function. The screening of thyroid function was very frequent in recent years, which might be due to the awareness of the association between thyroid function and kidney disease. Many AAV patients included in years much further back had no data of thyroid function, which made the sample size small. We expect to conduct a large-scale, multi-center collaborative study to investigate overall characteristics. Further investigation into the underlying mechanisms and potential treatment strategies may be a direction of future research.

## 5. Conclusions

In conclusion, thyroid dysfunction in AAV patients with renal injury was common, predominantly with hypothyroidism. Thyroid hormone levels were associated with renal function as well as renal outcomes in AAV patients with renal injury. The correlation between thyroid dysfunction and low hemoglobin levels in AAV patients with renal injury may provide a clue for the complex etiology of anemia in this population.

## Figures and Tables

**Figure 1 jpm-14-00099-f001:**
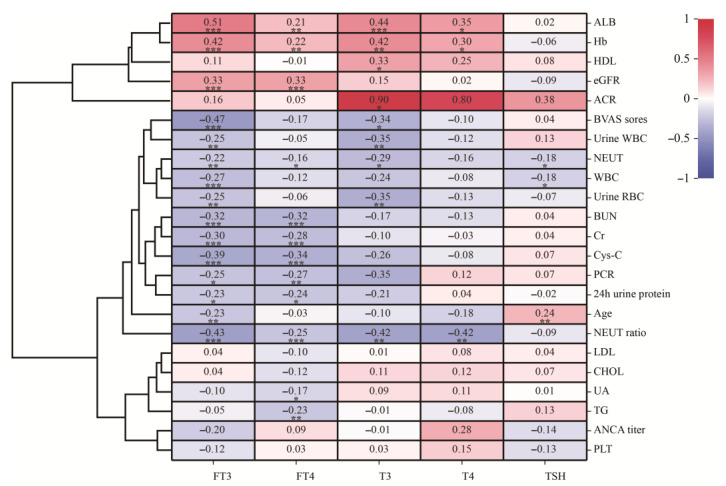
Heat map clustering of correlation coefficients between thyroid hormone and laboratory test in AAV patients with renal injury; * *p* < 0.05, ** *p* < 0.01, *** *p* < 0.001.

**Figure 2 jpm-14-00099-f002:**
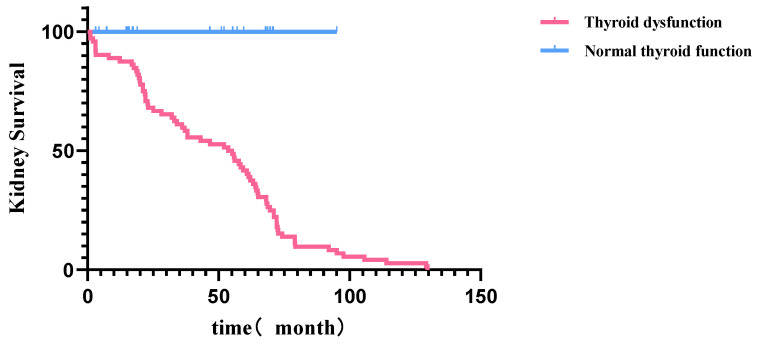
Kidney survival curve of AAV patients with renal injury in different groups.

**Table 1 jpm-14-00099-t001:** Comparisons of clinical parameters and laboratory findings by different groups. Data are expressed as mean ± standard deviation, median and interquartile range, or percent frequency, as appropriate. Hb, hemoglobin; NEUT ratio, neutrophil ratio; Alb, albumin; BUN, urea nitrogen; Cr, creatinine; BVAS, birmingham vasculitis activity index, version V3.0; PCR, urinary protein creatinine ratio; * *p* < 0.05.

	All (*n* = 174)	Normal Thyroid Group (*n* = 24)	Thyroid Dysfunction Group (*n* = 150)	*p*
Age (year)	56.9 ± 16.5	45.1 ± 16.2	58.8 ± 15.8	0.002 *
Elder (>60 years old, %)	84 (48.3%)	3 (12.5%)	81 (54.0%)	<0.001 *
Female (%)	102 (58.6%)	12 (50.0%)	90 (60.0%)	0.356
Hb (g/L)	88.41 ± 22.26	104.00 ± 24.93	85.92 ± 20.8	<0.001 *
NEUT ratio (%)	75.34 ± 11.77	68.37 ± 9.66	76.28 ± 11.43	0.002 *
Alb (g/L)	33.75 ± 6.42	38.86 ± 4.68	33.05 ± 6.28	0.003 *
BUN (mmol/L)	21.75 ± 15.20	12.50 ± 6.71	22.07 ± 13.15	<0.001 *
Cr (μmol/L)	431.02 ± 302.30	260.58 ± 215.55	457.14 ± 306.71	0.009 *
eGFR (mL/min/1.73 m^2^)	12.69 (6.96, 27.72)	44.32 (13.89, 68.75)	11.00 (6.62, 23.44)	<0.001 *
BVAS score	16.97 ± 6.22	13.67 ± 6.77	17.49 ± 5.98	0.005 *
FT3 (pmol/L)	2.66 ± 0.89	4.13 ± 0.40	2.45 ± 0.73	<0.001 *
FT4 (pmol/L)	14.02 ± 3.78	16.86 ± 2.25	13.58 ± 3.78	<0.001 *
TSH (mU/L)	2.66 (1.33, 4.89)	2.07 ± 1.27	2.80 (1.40, 5.45)	0.008 *
TT3 (nmol/L)	0.97 ± 0.39	1.55 ± 0.32	0.88 ± 0.32	<0.001 *
TT4 (nmol/L)	73.89 ± 24.40	101.93 ± 11.48	69.79 ± 23.08	0.003 *
PCR (g/mmol)	0.30 (0.17, 0.54)	0.251 ± 0.195	0.30 (0.17, 0.62)	0.091
24 h urinary protein (g)	1.61 (1.04, 3.52)	1.54 (0.79, 2.37)	1.94 (1.075, 4.065)	0.197
ACR (mg/mmol)	1944.70 ± 1675.55	1458.78 ± 1327.56	2033.05 ± 1742.79	0.539
Cys-C (mg/L)	3.89 ± 1.80	2.56 ± 1.46	4.11 ± 1.76	0.150
UA (μmol/L)	451.73 ± 143.00	441.92 ± 128.94	453.31 ± 145.18	0.678
TG (mmol/L)	1.66 ± 1.04	1.49 ± 0.91	1.66 ± 1.02	0.276
CHOL (mmol/L)	4.30 ± 1.58	4.10 ± 1.46	4.32 ± 1.53	0.335
HDL (mmol/L)	1.21 ± 0.53	1.16 ± 0.42	1.22 ± 0.55	0.565
LDL (mmol/L)	2.31 ± 1.09	2.24 ± 1.01	2.32 ± 1.06	0.535
PLT (×109)	225.29 ± 121.95	197.54 ± 85.24	231.14 ± 124.00	0.813
NEUT (×109)	7.20 ± 5.86	5.12 ± 2.08	7.46 ± 6.24	0.085

## Data Availability

All relevant data are within the manuscript.

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
