# Peer review of "Analysis of Thyroid Function in ANCA-Associated Vasculitis Patients with Renal Injury"

_jpm, 2024, doi:10.3390/jpm14010099_

Round 1

Reviewer 1 Report

Comments and Suggestions for Authors

The study aims to investigate the thyroid function in ANCA-associated vasculitis (AAV) patients with renal injury and explore the clinical relevance of renal dysfunction.

They performed a retrospective cohort study selecting AAV patients screened from 2011 to 2020 with data on thyroid function. Of 660 patients, only 174 had TSH, T3 and T4. Of these patients, 86.21% had thyroid dysfunction. At this point, two problems arise, the first one is about the adequacy of the sample size, and the second one is a possible bias since the tests might have been performed only in the patients with suspicion of thyroid dysfunction.

The main finding of the study was the association between kidney injury/insufficiency and thyroid dysfunction both in the regression model and in the kidney survival model. This association is well-known practically in CKD from any cause.

Author Response

1.Summary

2.

Does the introduction provide sufficient background and include all relevant references? ( ) (x) ( ) ( )
Are all the cited references relevant to the research? ( ) (x) ( ) ( )
Is the research design appropriate? ( ) (x) ( ) ( )
Are the methods adequately described? (x) ( ) ( ) ( )
Are the results clearly presented? (x) ( ) ( ) ( )
Are the conclusions supported by the results? ( ) ( ) (x) ( )

Response :The introduction section was resvised to be clearer about the research question and hypothesis.We also extended in the method section and the statistics section to enhance the reproducibility. In addition,the conclusions were also  modified  in the revised manuscript.

3.Point-by-point response to Comments and Suggestions for Authors

Comments 1: They performed a retrospective cohort study selecting AAV patients screened from 2011 to 2020 with data on thyroid function. Of 660 patients, only 174 had TSH, T3 and T4. Of these patients, 86.21% had thyroid dysfunction. At this point, two problems arise, the first one is about the adequacy of the sample size, and the second one is a possible bias since the tests might have been performed only in the patients with suspicion of thyroid dysfunction.

Response 1: Thank you very much for the comments. AAV is a relatively rare disease and the sample size was relatively small due to strict inclusion and exclusion criteria. In addition, in the past not all physician had the awareness of thyroid function screening.   The limitation of small sample size was described in the "limitation" section. For the possible bias that the included patients might be with suspicion of thyroid dysfunction, we checked all the medical records and found that all patients included had no history of previous thyroid disease. There were many AAV patients who did not have thyroid function data because different physician had different habits of screening for thyroid function. The screening of thyroid function was much frequent in recent years, which might be due to the awareness of the association between thyroid function and kidney disease.

Comments 2: The main finding of the study was the association between kidney injury/insufficiency and thyroid dysfunction both in the regression model and in the kidney survival model. This association is well-known practically in CKD from any cause.

Response 2: We appreciate for the great comment. It is true that accumulated research showed that thyroid dysfunction was common in patients with CKD. However, AAV patients with kidney injury was often manifested as rapidly progressive glomerulonephritis, with a rapid deterioration of kidney function in a short term. In the present study, we found a tight association between kidney injury/insufficiency and thyroid dysfunction in AAV patients, which supported a timely link between renal dysfunction and thyroid dysfunction. The similar association found in patients with AAV and those with CKD indicated that the renal impairment might be the key factor that induced the thyroid dysfunction in patients with kidney disease, which could also help explain the mechanism of thyroid dysfunction in patients with AAV.

Thank you for your careful review. We really appreciate your efforts in reviewing our manuscript during this unprecedented and challenging time. We wish good health to you, your family, and community. Your careful review has helped to make our study clearer and more comprehensive.

Reviewer 2 Report

Comments and Suggestions for Authors

The study investigates thyroid function in patients with ANCA-associated vasculitis (AAV) and renal injury. It included 174 patients without prior thyroid disease history. The findings revealed that out of the patients, 150 had thyroid dysfunction, predominantly hypothyroidism. This group differed in clinical parameters from those with normal thyroid function and were more likely to progress to dialysis dependence. Advanced age and decreased albumin were identified as independent risk factors for thyroid dysfunction. The study concludes that thyroid dysfunction, especially hypothyroidism, is common in AAV patients with renal injury and correlates with different clinical parameters and a higher progression to dialysis dependence.

minor clarification and suggestions 

The article could be improved by refining the introduction to more clearly state the research question and hypothesis. A more detailed methodology section, particularly regarding patient selection and data collection methods, would enhance reproducibility. The statistical analysis section needs expansion, explaining the choice and application of statistical tests. Results should be interpreted more deeply in the context of existing literature. Improving the clarity and labeling of graphs and tables would aid in data visualization. Discussing the study’s limitations, including its retrospective nature, more comprehensively, and detailing the ethical considerations are crucial. Updating references to ensure they are current and relevant, providing a more specific conclusion that summarizes findings and implications, and offering clear directions for future research based on the study’s outcomes would significantly enhance the article’s scientific value and clarity.

Author Response

 1.Summary

2.

Does the introduction provide sufficient background and include all relevant references? (x) ( ) ( ) ( )
Are all the cited references relevant to the research? (x) ( ) ( ) ( )
Is the research design appropriate? ( ) (x) ( ) ( )
Are the methods adequately described? ( ) (x) ( ) ( )
Are the results clearly presented? ( ) (x) ( ) ( )
Are the conclusions supported by the results? ( ) (x) ( ) ( )

Response :The introduction section was resvised to be clearer about the research question and hypothesis.We also extended in the method section and the statistics section to enhance the reproducibility. In addition,the conclusions were also  modified  in the revised manuscript.

3.Point-by-point response to Comments and Suggestions for Authors

Comments 1: The article could be improved by refining the introduction to more clearly state the research question and hypothesis. A more detailed methodology section, particularly regarding patient selection and data collection methods, would enhance reproducibility. The statistical analysis section needs expansion, explaining the choice and application of statistical tests. Results should be interpreted more deeply in the context of existing literature. Improving the clarity and labeling of graphs and tables would aid in data visualization. Discussing the study’s limitations, including its retrospective nature, more comprehensively, and detailing the ethical considerations are crucial. Updating references to ensure they are current and relevant, providing a more specific conclusion that summarizes findings and implications, and offering clear directions for future research based on the study’s outcomes would significantly enhance the article’s scientific value and clarity.

Response 1Thank you very much for the suggestions. We have revised the introduction section to be clearer about the research question and hypothesis. And many details were extended in the method section and the statistics section to enhance the reproducibility. The clarity and labeling of graphs and tables was improved in revised manuscript. In addition, analysis and references were added based on the results. The limitation part has also been improved and expanded. The references have been updated to be more current and relevant. Conclusion was improved to be more specific. We pointed out the clear possible research direction in the future based on the study’s outcomes in revised manuscript.

Thank you for your careful review. We really appreciate your efforts in reviewing our manuscript during this unprecedented and challenging time. We wish good health to you, your family, and community. Your careful review has helped to make our study clearer and more comprehensive.

Reviewer 3 Report

Comments and Suggestions for Authors

Issues to be addressed: 

1. In methodology authors stated that the study was a retrospective analysis. However,  merian follow up of 68 months was also mentioned in the text. Clarify please,  is it a retrospective or prospective study?

2. Again in methodology,  authors stated different comparison tests for variables with normal and skewed distribution. However  it is not clear which normality test was conducted. 

3. Discussion must be more than speculation and repetition of the results.improve it please. Moreover, clinical utility of the findings should be commented on.

Comments on the Quality of English Language

Minor spelling errors could be fixed up

Author Response

 1. Summary

2.

Does the introduction provide sufficient background and include all relevant references? (x) ( ) ( )   ( )
Are all the cited references relevant to the research? (x) ( ) ( )   ( )
Is the research design appropriate? ( ) (x) ( )   ( )
Are the methods adequately described? ( ) (x) ( )   ( )
Are the results clearly presented? (x) ( ) ( )   ( )

Are the conclusions supported by the results? ( ) (x) ( )   ( )

Response and Revisions:The introduction section was resvised to be clearer about the research question and hypothesis.We also extended in the method section and the statistics section to enhance the reproducibility. In addition,the conclusions were also  modified  in the revised manuscript.

3. Point-by-point response to Comments and Suggestions for Authors

Comments 1:  In methodology authors stated that the study was a retrospective analysis. However,  merian follow up of 68 months was also mentioned in the text. Clarify please,  is it a retrospective or prospective study?

Response 1: Thank you very much for the comments. This is a retrospective and prospective study. We retrospectively included clinical data from 174 AAV patients and then conducted follow-up to collect data of prognosis. In the survival analysis, the exposure factor was thyroid dysfunction, and the endpoint was dialysis dependence or death. Therefore, we consider this study to be a retrospective and prospective study. The statement in the manuscript was modified accordingly.

Comments 2: Again in methodology,  authors stated different comparison tests for variables with normal and skewed distribution. However  it is not clear which normality test was conducted. 

Response 2: Thank you very much. The methods of Kolmogorov-Smirnov test and P-P diagram were used for normality test. We have included this information in “statistics” section in revised manuscript.

Comments 3:Discussion must be more than speculation and repetition of the results.improve it please. Moreover, clinical utility of the findings should be commented on. 

Response 3:Thanks for the suggestion. The “discussion” section has been expanded based on the results. Clinical utility of the findings was also expanded in revised manuscript.

4. Response to Comments on the Quality of English Language

Response :The revised manuscript has been edited with the help of a native English speaker.

Thank you for your careful review. We really appreciate your efforts in reviewing our manuscript during this unprecedented and challenging time. We wish good health to you, your family, and community. Your careful review has helped to make our study clearer and more comprehensive.

Round 2

Reviewer 1 Report

Comments and Suggestions for Authors

The paper has not substantially changed. I think that it offers an interesting clinical observation susceptible to future in-depth analysis, but the main flaws cannot be addressed:

- of 636 patients 472 had no thyroid function data,

- of 174 patients, only 24 had a normal thyroid function.

This study may be the starting point for a more structured one.

Author Response

   Thank you very much for taking the time to review this manuscript. Please find the detailed responses below and the corresponding revisions in the resubmitted files. 

Comments 1: The paper has not substantially changed. I think that it offers an interesting clinical observation susceptible to future in-depth analysis, but the main flaws cannot be addressed:

- of 636 patients 472 had no thyroid function data,

- of 174 patients, only 24 had a normal thyroid function.

This study may be the starting point for a more structured one.

Response 1: Thank you very much for the comments. The data for this study was collected retrospectively from clinical databases, there were many AAV patients who did not have thyroid function data because different physician had different habits of screening for thyroid function. The screening of thyroid function was much frequent in recent years, which might be due to the awareness of the association between thyroid function and kidney disease. Many AAV patients included in the much further back years had no data of thyroid function made the sample size be small. The screening for thyroid function was decided by physicians. There might be possible bias that the included patients might be with suspicion of thyroid dysfunction. We checked all the medical records and found that all patients included had no history of previous thyroid disease. However, the bias could not be completely avoided. The limitation of the study was described in detail in limitations part of the discussion section in the revised manuscript.

   Of the 174 patients, only 24 had a normal thyroid function. In the present study, the thyroid dysfunction was defined and classified based only on hormone levels and was not equivalent to clinical diagnosis (stated in the method section). The relatively loose definition might contribute to the larger size in the thyroid dysfunction group. In the present study, we had found 36.7% patients with hypothyroidism. Wiseman P, et al had reported that hypothyroidism occurred in 30% of patients with giant cell arteritis, in 56% of patients with polymyalgia rheumatica (BMJ 1989;298:647–8.). In a recent study by Kermani T A, et al. (Rheumatology (Oxford), 2022, 61(7): 2942-2950.), hypothyroidism was found to be present in 10% of patients with vasculitis. The highest frequency of hypothyroidism was in patients with MPA (18%), a subtype of ANCA associated vasculitis. In the present study, the main patients included were MPA, which might explain the relatively higher rate of thyroid dysfunction.

   The present study revealed that thyroid dysfunction was common in patients with AAV, which might help reinforce the awareness of the association between thyroid function and kidney disease for nephrologist. It is true that the present study was a very preliminary study. A more structured one was needed. Since AAV is a relatively rare disease, accumulating clinical data of newly diagnosed patients may take a long time. We look forward to the corresponding research in the future.